# Identification and prevalence of frailty in diabetes mellitus and association with clinical outcomes: a systematic review protocol

Peter Hanlon ⬤ , Isabella Fauré, Neave Corcoran, Elaine Butterly, Jim Lewsey, David A McAllister, Frances S Mair

DAM and FSM are joint senior authors.

Institute of Health and Wellbeing, University of Glasgow, Glasgow, UK

**Correspondence to**
Dr Peter Hanlon;
peter.hanlon@glasgow.ac.uk

## ABSTRACT

**Introduction** Diabetes mellitus is common and growing in prevalence, and an increasing proportion of people with diabetes are living to older age. Frailty is, therefore, becoming an important concept in diabetes. Frailty is associated with older age and describes a state of increased susceptibility to decompensation in response to physiological stress. A range of measures have been used to quantify frailty. This systematic review aims to identify measures used to quantify frailty in people with diabetes (any type); to summarise the prevalence of frailty in diabetes; and to describe the relationship between frailty and adverse clinical outcomes in people with diabetes.

**Methods and analysis** Three electronic databases (Medline, Embase and Web of Science) will be searched from 2000 to November 2019 and supplemented by citation searching of relevant articles and hand searching of reference lists. Two reviewers will independently review titles, abstracts and full texts. Inclusion criteria include: (1) adults with any type of diabetes mellitus; (2) quantify frailty using any validated frailty measure; (3) report the prevalence of frailty and/or the association between frailty and clinical outcomes in people with diabetes; (4) studies that assess generic (eg, mortality, hospital admission and falls) or diabetes-specific outcomes (eg, hypoglycaemic episodes, cardiovascular events, diabetic nephropathy and diabetic retinopathy); (5) cross-sectional and longitudinal observational studies. Study quality will be assessed using the Newcastle–Ottawa Scale for observational studies. Clinical and methodological heterogeneity will be assessed, and a random effects meta-analysis performed if appropriate. Otherwise, a narrative synthesis will be performed.

**Ethics and dissemination** This manuscript describes the protocol for a systematic review of observational studies and does not require ethical approval.

**PROSPERO registration number** CRD42020163109.

## Strengths and limitations of this study

► This systematic review will provide a comprehensive overview of the prevalence and implications of frailty in people with diabetes.
► We will include a broad range of frailty definitions and clinical outcomes relevant to diabetes.
► There is likely to be significant heterogeneity between population characteristics and frailty definitions in included studies.
► By including only English language articles, there is a chance of language bias in the results of the review.
► We exclude Grey literature, which may lead to publication bias.

resulting from destruction of pancreatic beta cells, usually by an autoimmune process.[2] Type 2 diabetes describes a relative insulin deficiency caused by beta-cell dysfunction and insulin resistance of target organs.[2] Both are associated with a range of complications including macrovascular disease, retinopathy, nephropathy and neuropathy.[3] The prevalence of diabetes is increasing across the world.[4] Population demographics are also shifting towards an ageing population.[5] Among people above the age of 65, the prevalence of diabetes can be as high as 30%.[6] Diabetes in older people is, therefore, a growing clinical and public health priority. One factor with important implications for disease management in older age is frailty.[7]

Frailty is a state characterised by reduced functional reserve across multiple physiological systems.[8] People living with frailty have impaired resolution of homaeostasis following physiological stressors.[8] Frailty, therefore, carries an increased risk of a range of adverse health outcomes, such as falls, cognitive decline, hospital admission and mortality.[9] Frailty is widely recognised

## INTRODUCTION

Diabetes mellitus (hereafter 'diabetes') describes a collection of metabolic disorders, with distinct pathological processes, that are characterised by elevated blood glucose.[1] The most common are type 1 and type 2 diabetes. Type 1 diabetes is caused by insulin deficiency

to be a multidimensional and dynamic state, associated with older age and with a range of non-communicable diseases.[9] However, there is no single universally accepted operational definition of frailty. Rather, a wide range of definitions have been used in both research and clinical practices.[10]

The two dominant paradigms in the frailty literature are the frailty phenotype and the frailty index. The frailty phenotype, described by Fried *et al*, defines frailty as the presence of three or more out of five features: low hand grip strength, unintentional weight loss, low physical activity, exhaustion and slow walking pace.[11] The presence of one or two of these features is classified as a prefrail state. The frailty index, described by Rockwood and Mitnitski, is based on a Cumulative Deficit Model of frailty, whereby frailty is identified by counting the number of health 'deficits' present in an individual.[12] At least 30 deficits are required to construct a frailty index, all of which must increase in prevalence with age, be associated with poor health and not saturate too early (ie, be universally present among older people).[13] Both the frailty phenotype and the frailty index have been associated with adverse health outcomes in a range of older populations; however, the populations identified as frail by each are different.[14] Since their original description, a wide range of other frailty instruments, as well as adaptations of the frailty index and phenotype, have been developed for both epidemiological studies and for clinical practice.[9 10]

The relationship between diabetes and frailty is complex. Diabetes is associated with a higher prevalence of frailty.[15–18] Both type 1 and type 2 diabetes lead to microvascular and macrovascular complications that have important physical, cognitive and functional consequences, which may contribute to the development of frailty.[6] Hyperglycaemia is also recognised to directly impact muscle mass and quality, exacerbating age-related sarcopaenia and, in turn, physical function.[19] However, the association between frailty and poor functional outcomes in people with diabetes is only partially explained by direct complications of diabetes.[17 20]

The importance of frailty in the context of diabetes is increasingly recognised in clinical guidelines.[7] Specifically, higher glycated haemoglobin (HbA1c) targets are recommended in the context of frailty, in part due to the increased risks associated with hypoglycaemia.[21] Despite this, up to 40% of older people with diabetes may be overtreated (with HbA1c<7%).[22 23] Conversely, poor glycaemic control and associated vascular complications risk causing, or accelerating the progression of, frailty.[24]

One recent meta-analysis demonstrated a consistent relationship between frailty and mortality, hospitalisation and cardiovascular events in the context of diabetes.[25] We are not aware of any systematic review to assess the prevalence of frailty in diabetes, or to consider a broader range of outcomes relevant to the management of diabetes.

To enhance understanding of the implications and management of diabetes within an ageing population,

it is important to fully describe the association between diabetes and frailty. Given the risks of both over treatment and under treatment of diabetes in the context of frailty, it is important to understand the associations between frailty and a range of potential outcomes in diabetes. This includes generic outcomes such as mortality and hospitalisation and disability and disease-specific outcomes such as retinopathy, neuropathy and hypoglycaemic events. An understanding of the range and complexity of these associations is required to inform clinical decisions around treatment priorities and to underpin future research. This includes quantifying the prevalence of frailty in people with diabetes, and the impact that different frailty definitions might have on this prevalence. This manuscript describes the protocol of a systematic review aiming to synthesise existing evidence relating to these questions.

## Aims

The systematic review will aim to:

▶ Identify which frailty measures have been used to assess frailty in people with diabetes (any type, including mixed/unspecified).

▶ Quantify the prevalence of frailty among people with diabetes.

▶ Describe the association between frailty and both generic (eg, mortality) and disease-specific (eg, hypoglycaemia) clinical outcomes in the context of diabetes.

## METHODS AND ANALYSIS

The review will be conducted and reported according to the Preferred Reporting Items for Systematic Reviews and Meta-Analyses statement.[26] Where a meta-analysis is undertaken, we will report findings according to the Meta-analyses Of Observational Studies in Epidemiology checklist.

### Eligibility criteria for inclusion

The eligibility criteria for this review are summarised in table 1 and explained in more detail below.

### Population

We will include studies analysing data from people with any form of diabetes.

While frailty is a state associated with increasing age, there is evidence that frailty is identifiable in relatively younger people, particularly in certain contexts such as multimorbidity (two or more coexisting long-term conditions) or in areas of high socioeconomic deprivation. We will, therefore, include studies of adults of any age (≥18 years). However, we anticipate that most studies will focus predominantly on 'older' populations.

From an initial scoping of the literature, it is likely that many studies describing frailty in population-based studies measure unspecified 'diabetes' rather than explicitly type 1 or type 2 diabetes. We will, therefore, include any study that includes people with any type of diabetes (including type 1, type 2 diabetes, secondary or monogenic diabetes, or people with unspecified diabetes). Given that frailty is

| Table 1 | Inclusion criteria |
|---|---|
| **PECOS component** | **Description** |
| Population | Adults (≥18 years old)<br>Diabetes (any type, including mixed or unspecified) |
| Exposure | Frailty as assessed by any frailty measure |
| Comparator | People with diabetes not classified as frail |
| Outcomes | Generic:<br>► Mortality<br>► Major adverse cardiovascular events<br>► Hospital admission<br>► Admission to long-term care facility<br>► Falls<br>► Number of clinic attendances<br>► Quality of life<br>► Disability/functional status<br><br>Diabetes specific:<br>► HbA1c (cross-sectional association, or longitudinal)<br>► Glycaemic variability<br>► Hypoglycaemic episodes<br>► Diabetic retinopathy (cross-sectional association, or longitudinal)<br>► Diabetic nephropathy (cross-sectional association, or longitudinal)<br> – Include development of end-stage renal disease<br>► Diabetic foot complications (cross-sectional association or longitudinal)<br>► Treatment burden (eg, Diabetic Treatment Burden Questionnaire) |
| Settings | Community (including care home/nursing home)<br>Outpatient clinic<br>Inpatient |
| Study design | Cross-sectional or longitudinal<br>Cohort |
| Other exclusions | Conference abstracts, letters, review articles, intervention studies and Grey literature |

PECOS, Population, Exposure, Comparator, Outcome, Setting and Study design.

a state associated with older age, and that type 2 diabetes is both more prevalent than type 1 diabetes and becomes more prevalent with age, it is likely that most (but not all) people with diabetes in the relevant populations will have type 2 diabetes. Studies of type 1, type 2 diabetes and those of unspecified diabetes will be considered separately in any subsequent analysis.

We will include studies focusing purely on people with diabetes, or population-based studies that report results for people with diabetes separately.

### Exposure

The 'exposure' of interest is frailty. Many epidemiological measures and clinical tools have been developed to identify frailty for research or clinical practice.[10]

To be eligible for inclusion, a study must use a measure that explicitly seeks to quantify frailty. We will include measures developed primarily as epidemiological tools (eg, the frailty phenotype and frailty index).[11 12] We will also include measures designed primarily for clinical practice (eg, the Clinical Frailty Scale).[27]

Studies focusing solely on comorbidity (ie, no additional measures to identify 'frailty') will be excluded unless these are explicitly operationalised as a 'frailty index'. In this case, studies would generally be expected to include additional deficits (such as symptoms, functional limitations and laboratory measures). Studies that use a single parameter as a proxy for frailty (eg, grip strength alone and self-rated health) will be excluded.

### Comparator

Studies that report the prevalence of frailty will be eligible for inclusion if they report the prevalence of frailty in diabetes only. Studies should report the number or proportion of participants with and without frailty (or with varying degrees of frailty, depending on the measure used).

For assessing the association between frailty and clinical outcomes in the context of diabetes, studies should report the association between frailty and the outcome of interest. This may be reported either as the association with the presence or absence of frailty (in the case of a

## Box 1 Medline search

1. Exp Frailty/
2. Exp Frail Elderly/
3. Frail*.tw
4. 1 or 2 or 3
5. Exp Diabetes Mellitus
6. Diabet*.tw
7. (IDDM or NIDDM or MODY or T1DM, or T2DM or T1D or T2D).tw
8. (non insulin* depend* or non insulin depend* or non insulin?depend* or non insulin ?depend).tw
9. (insulin* depend* or insulin ?depend*).tw
10. 5 or 6 or 7 or 8 or 9
11. Exp Diabetes Insipidus/
12. Diabet* insipidus.tw
13. 11 or 12
14. 10 not 13
15. 4 and 14

binary or categorical measure) or the association between the degree of frailty and the outcome (in the case of a continuous or ordinal measure of frailty).

### Outcomes

Outcomes of interest are summarised in table 1. We will include studies assessing any of these outcomes as long as the association is specifically quantified in people with diabetes and frailty.

### Setting

We will include studies of community-dwelling patients, outpatient populations or hospital inpatients.

For the purposes of this review, given the focus on frailty, people living in long-term care facilities (eg, care homes and nursing homes) will be considered to be 'community dwelling'. Therefore, any study including, or specifically recruiting, nursing home residents will be eligible for inclusion.

### Identification of studies
#### Electronic searches

Medline, Embase and Web of Science (core collection) databases will be search using a combination of Medical Subject Headings and keyword searches (online supplementary file 1). The terms used for the Medline search are shown in box 1. These terms will be adapted for the other databases. Searches will be from 2000 to November 2019. The year 2000 was chosen as the start date as the first seminal paper operationalising the concept of frailty in an epidemiological study was published in 2001. Articles published prior to this date are, therefore, unlikely to be relevant. No language restriction will be applied to the search, but only English language articles will be included at the screening level. This language restriction is a pragmatic decision; however, we acknowledge that this may lead to a language bias in the results, potentially excluding relevant studies published in other languages.

### Identifying additional articles

Electronic searches will be supplemented by hand searching reference lists of relevant articles. A citation search of all relevant articles will also be carried out using the Web of Science citation search tool.

### Data collection and analysis
#### Selection of studies

Two reviewers, working independently, will screen all titles and abstracts of records identified in the database searches. PECOS (population, exposure, comparator, outcome, setting and study design) criteria outlined above will be used to determine eligibility. Where there is disagreement, studies will be retained for full-text screening.

Full texts of all potentially eligible studies will be screened independently by two reviewers. Disagreements about eligibility will be resolved by consensus, involving a third reviewer where necessary.

#### Data extraction

A standard data extraction form will be designed and piloted before being applied to each of the included studies. Extracted data will include:

Study details
► Author.
► Year
► Location.
► Setting (community, outpatient and residential care).
► Method of recruitment (eg, random sample, postal invitation and consecutive patients).
► Method of assessment (face to face, survey and linkage to healthcare records).

Population
► Age.
► Sex.
► Ethnicity.
► Socioeconomic status.
► Comorbidities.
► Medications.
► Social circumstances (eg, living independently, requiring carers, family support and so on).
► Smoking status.
► Physical activity.

Diabetes details
► Type of diabetes.
► Method of confirmation (self-report, medical records and clinical assessment).
► Measure of control (eg, HbA1c).
► Medication (eg, proportion taking insulin, oral antidiabetics and so on).
► Presence and severity of complications (eg, retinopathy, nephropathy, neuropathy, ulceration and Charcot arthropathy).

Frailty definition
► Frailty measure used.
► Definitions for each component of the frailty measure (eg, cut-points used for continuous measures and

method of assessment (questionnaire, interview and so on)).
Frailty prevalence
Outcomes (generic):

► Mortality.
► Major adverse cardiovascular events.
► Hospital admission.
► Admission to long-term care facility.
► Falls.
► Number of clinic attendances.
► Quality of life.
► Disability/functional status

Outcomes (diabetes specific):

► HbA1c (cross-sectional association or longitudinal).
► Glycaemic variability.
► Hypoglycaemic episodes.
► Diabetic retinopathy (cross-sectional association or longitudinal).
► Diabetic nephropathy (cross-sectional association or longitudinal).
► Diabetic foot complications (cross-sectional association or longitudinal).
► Treatment burden (eg, Diabetic Treatment Burden Questionnaire).

As we include a wide range of outcomes, it is likely that the way outcomes are assessed will vary depending on the outcome in question. Studies may also assess similar outcomes (eg, hospital admission) in different ways (eg, number of admissions over specified follow-up, time to first admission and presence or absence of admission during follow-up). For the outcomes listed above, we will extract data regardless of the method of assessment. Heterogeneity in the way outcome data were collected will be used to inform the approach to data synthesis (ie, meta-analysis vs narrative synthesis). For each outcome reported, we will record:

► The method of outcome assessment (eg, linkage to healthcare records, face-to-face assessment, questionnaire and so on).
► Method of analysis (eg, time to event, mean difference and so on).
► The association between frailty and the outcome (eg, prevalence, OR, HR and so on).
► Adjustment for any potential confounders.
► Length of follow-up over which the outcome was assessed.
► Method of analysis of competing risks when assessing each outcome.

Where available, we will also extract data on both relative (eg, HRs) and absolute (eg, events per 1000 people) associations with outcomes.

## Assessment of methodological quality

The Newcastle–Ottawa Scale will be used to assess the risk of bias for each study (online supplementary file 2).[28] This scale is widely used for the assessment of observational studies, and has frequently been adapted to the context of specific systematic reviews. We have adapted the criteria

in order to be explicit about how the 'exposure assessment' related to frailty: specifically, awarding one point for the use of a validated frailty assessment measure. For cross-sectional studies, only the first five elements of the scale were relevant to quality assessment (the remainder concerning the longitudinal assessment of outcomes). We will use this subsection of the Newcastle–Ottawa Scale to assess the quality of cross-sectional studies to allow direct comparability with the baseline assessments of longitudinal studies (from which we will also extract data on frailty prevalence). In assessing the comparability of frail/non-frail groups, age will be taken as the most important factor for which studies should account.

## Data synthesis

The appropriate method of data synthesis will be determined after assessment of the heterogeneity of the included studies, in terms of population selection and demographics, frailty definition and method of outcome assessment.

With regards to the prevalence of frailty, different frailty measures will be considered separately (ie, we will not perform a meta-analysis of frailty prevalence measured using different scales). We will also consider community studies separately from studies focussing on outpatient clinic populations (as these may represent people with more severe diabetes), inpatients or people living in residential care. We will also assess the inclusion criteria and demographics of the sample population, with particular attention to age (as frailty is strongly associated with age) and sex (as women tend to have a higher prevalence of frailty than men) to determine the most appropriate method of synthesis. Where samples have been drawn from populations with a markedly different age/sex structure, a pooled estimate of the mean prevalence of frailty across these studies is unlikely to be a meaningful summary. Similarly, other inclusion criteria used by the individual studies (such as excluding 'institutionalised' people, people with cognitive impairment and people with impaired mobility unable to attend an assessment) may disproportionately impact on the estimation of frailty prevalence. The appropriateness, or otherwise, of a meta-analysis of frailty prevalence will be judged only after examination of these aspects of the included studies.

For the assessment of outcomes, the approach to synthesis will also be judged based on heterogeneity of the method of outcome assessment and the analytic approach. As above, different frailty measures will be considered separately.

If appropriate, we will combine these in a random effects meta-analysis (anticipating heterogeneity in the true association). As well as a pooled estimate and 95% CIs, we will also calculate the prediction interval to assess the range of plausible estimates from the observed data. Heterogeneity will be quantified using the $I^2$ statistic. Where heterogeneity is present, we will attempt to explore potential sources of heterogeneity using subgroup analyses (eg, by method of determining frailty, age of sample population

and method of outcome assessment). By doing so, we propose to explore factors that may influence the estimates reported in observational studies in the presence of heterogeneity, rather than provide a definitive single estimate.[29] We will use funnel plots to assess for potential publication bias.

Only those studies that are judged to be sufficiently comparable will be included in meta-analyses. For outcomes where there are too few studies, or the included studies are too heterogenous to permit a meaningful meta-analysis (eg, in terms of outcome definition or method of assessing frailty), we will perform a narrative synthesis of the study findings. This will report the methods used to identify frailty along with the prevalence and association with outcomes, to explore the impact of the method of assessment on the observed relationship. This will be reported alongside detail of the recruitment strategy, age profile and characteristics of each sample included.

### Patient and public involvement

No patients were involved in the development of this review.

## ETHICS AND DISSEMINATION

This systematic review will provide an overview of the prevalence of frailty in diabetes and the relationship between frailty and adverse health outcomes in people with diabetes.

As the prevalence of both frailty and diabetes increase, it will become increasingly important for clinical guidelines for the treatment of diabetes to explicitly consider the needs of people living with frailty.[7] Quantifying the prevalence of frailty in diabetes will allow the scale of this challenge to be better appreciated. By including any reported definition of frailty within our inclusion criteria, this review will demonstrate which of the wide range of frailty instruments and measures have been used to study frailty in diabetes. It will also be possible to compare if and how prevalence and association with outcomes differs depending on the frailty definition used.

Given the likely heterogeneity in frailty definitions, as well as inherent differences in the populations studied, it may not be possible to undertake a meta-analysis of the findings of this review. If this is the case, we propose to conduct a detailed narrative synthesis, systematically describing and synthesising details of the populations under study as well as the details of frailty definitions used.

We also propose to search for and extract data for a wide range of clinical outcomes. Given the multidimensional nature of frailty,[8] and the vulnerability to decompensation that is inherent to any frailty definition,[9] it is likely that frailty will be associated with a range of adverse outcomes. The challenge in translating these associations into meaningful recommendations is understanding the balance of these risks, and how they might inform clinical decisions and recommendations. The balance of risks in

diabetes, and treatment priorities, may differ depending on the degree of frailty experienced by an individual. The associations may also differ in their nature or magnitude depending on the method used to identify frailty. This review will aim to provide an overview of what is known about the relationship between frailty and both generic and disease-specific outcomes. This is likely to inform priorities for future research into the consequences of frailty in diabetes.

As this project is a systematic review, ethical approval is not required. Patients or the public were not involved in the development of this protocol.

**Contributors** All authors contributed to the conception and design of the proposed study. PH, DM and FM developed the data sources and search strategy. PH, IF, NC, DM and FM refined the inclusion criteria. PH, IF, NC, DM and FM developed the data extraction template which was piloted by PH, IF and NC. PH and IF wrote the first draft. All authors critically reviewed this and subsequent drafts. All authors approved the final version of the manuscript for submission. FM is the guarantor of the review. All authors accept accountability for the accuracy of the protocol.

**Funding** PH was funded by a Medical Research Council Clinical Research Training Fellowship (grant reference MR/S021949/1) entitled Understanding the prevalence and impact of frailty in chronic disease and implications for clinical management.

**Competing interests** None declared.

**Patient and public involvement** Patients and/or the public were not involved in the design, or conduct, or reporting, or dissemination plans of this research.

**Patient consent for publication** Not required.

**Provenance and peer review** Not commissioned; externally peer reviewed.

**ORCID iD**
Peter Hanlon http://orcid.org/0000-0002-5828-3934

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
