## [Reviewer comments · BMJ Open]

ARTICLE DETAILS

TITLE (PROVISIONAL)	The identification and prevalence of frailty in diabetes mellitus and association with clinical outcomes: A systematic review protocol
AUTHORS	Hanlon, Peter; Fauré, Isabella; Corcoran, Neave; Butterly, Elaine; Lewsey, Jim; McAllister, David; Mair, Frances

VERSION 1 – REVIEW

REVIEWER	Nanna Lindekilde Department of Psychology, University of Southern Denmark Denmark
REVIEW RETURNED	20-Feb-2020

GENERAL COMMENTS	Thank you very much for the opportunity to read and review your protocol for a systematic review on the prevalence of frailty in diabetes mellitus and association with clinical outcomes. I have listed my comments and questions below. Furthermore, I have marked and commented a few specific places in the document, see attached. 1) General: The authors switch between using the terms “diabetes” and “diabetes mellitus” throughout the whole paper. This should be streamlined. 2) The title: In the paper there is three aims; the measurements, the prevalence and association to clinical outcome. However, in the title there is only focus on the prevalence and association with outcome which I found unfortunate. 3) The introduction: there need information about diabetes – e.g. definition and description (e.g. subtypes) in the beginning. It is clear that the paper is written with focus on frailty, however, to describe frailty in association to diabetes, a description of diabetes is needed. I would recommend including this in the first section of the introduction. 4) The study population: It is unclear to me, what age the systematic review will focus on. Some section “older people” are mentioned but without a definition and other sections it says > 18 years. 5) Table 1. In exposure it says that it must be a validated measurement. This demand of the measurement being validated is not mentioned elsewhere. I think it would be a mistake to have this as a demand. Many measures used are not validated or is validated very poorly. Regarding the focus in the paper, I would also think it is relevant to include not-validated measurements if they are used.
--

	6) References: There is a lack of references. Mange associations are described without any references to support it.
REVIEWER	Paulo Ricardo Saquete Martins-Filho Investigative Pathology Laboratory, Federal University of Sergipe, Brazil
REVIEW RETURNED	24-Feb-2020
GENERAL COMMENTS	Thank you for considering me as a reviewer for this publication in your esteemed journal. I have provided my comments as follows.  1. Is it possible to use the PECO (Population, Exposure, Comparator and Outcomes) model to define eligibility criteria, since observational studies will be included in the search ? 2. Why did the authors use PRISMA statement instead of MOOSE statement to develop this protocol? 3. Page 10: "studies should report the association with the outcome in the presence of absence of frailty". I think the authors would like to say "in the presence or absence of frailty". 4. I think the authors should explain in more detail how the outcomes should be measured. 5. Authors should review in their protocol a possible language bias. 6. Authors should include gray literature in the search. 7. The methodological assessment of cross-sectional studies should be performed using another tool. See https://www.nhlbi.nih.gov/health-topics/study-quality-assessment-tools 8. The strategies for planning the meta-analysis are vague. Please describe in detail how the measures of proportion and association can be summarized. Will there be sensitivity analysis and publication bias analysis?

VERSION 1 – AUTHOR RESPONSE

Reviewer 1

1) General: The authors switch between using the terms “diabetes” and “diabetes mellitus” throughout the whole paper. This should be streamlined.

Author response

We thank the reviewer for pointing this out. We have altered the manuscript to use the term “diabetes mellitus (hereafter diabetes)” at the first mention (in the new section defining the term, page 4, line 70) and for the remainder of the manuscript use the term “diabetes”.

2) The title: In the paper there is three aims; the measurements, the prevalence and association to clinical outcome. However, in the title there is only focus on the prevalence and association with outcome which I found unfortunate.

Author response

Thank you for this comment. We have altered the title which now reads: “The identification and prevalence of frailty in diabetes mellitus and association with clinical outcomes: A systematic review protocol”.

3) The introduction: there need information about diabetes – e.g. definition and description (e.g. subtypes) in the beginning. It is clear that the paper is written with focus on frailty, however, to describe frailty in association to diabetes, a description of diabetes is needed. I would recommend including this in the first section of the introduction.

Author response

Thank you for the suggestion. We have added an opening paragraph to the manuscript defining diabetes and introducing type 1 and type 2 diabetes as the two most common entities. This paragraph states:

“Diabetes mellitus (hereafter “diabetes”) describes a collection of metabolic disorders, with distinct pathological processes, that are characterised by elevated blood glucose. The most common are type 1 diabetes and type 2 diabetes. Type 1 diabetes is caused by insulin deficiency resulting from destruction of pancreatic beta cells, usually by an autoimmune process. Type 2 diabetes describes a relative insulin deficiency caused by beta-cell dysfunction and insulin resistance of target organs. Both are associated with a range of complications including macrovascular disease, retinopathy, nephropathy and neuropathy. “ Page 4 lines 70-76

4) The study population: It is unclear to me, what age the systematic review will focus on. Some section “older people” are mentioned but without a definition and other sections it says > 18 years.

Author response

Thank you for highlighting the need for clarification. Our review will focus on adults of any age (e.g. >18 years). We have taken this approach as frailty has been identified in middle-aged as well as older adults, particularly in certain circumstances such as multimorbidity, high socioeconomic deprivation, people living with homelessness etc. We therefore did not wish to limit our inclusion criteria by an arbitrary age cut-off as capturing this literature would be important.

We have added the following to the ‘population’ section to clarify this:

“While frailty is a state associated with increasing age, there is evidence that frailty is identifiable in relatively younger people, particularly in certain contexts such as multimorbidity (2 or more co-existing long-term conditions) or in areas of high socioeconomic deprivation. We will therefore include studies of adults of any age (≥ 18 years). However, we anticipate that most studies will focus predominantly on ‘older’ populations.” Page 8, lines 147-151

5) Table 1. In exposure it says that it must be a validated measurement. This demand of the measurement being validated is not mentioned elsewhere. I think it would be a mistake to have this as a demand. Many measures used are not validated or is validated very poorly. Regarding the focus in the paper, I would also think it is relevant to include not-validated measurements if they are used.

Author response

Thank you for this comment. We agree with the reviewer and have removed the criteria for a ‘validated’ measure from the inclusion criteria (table 1, page 7). We will assess whether or not a measure was validated at the quality assessment stage, but will not exclude papers on this basis (for the reasons the reviewer states: namely that many measured may not be, or be poorly, validated).

6) References: There is a lack of references. Many associations are described without any references to support it.

Author response

We have carefully revised the manuscript with particular attention to the introduction and discussion

section in order to add references to support all statements made.

Reviewer: 2

1. Is it possible to use the PECO (Population, Exposure, Comparator and Outcomes) model to define eligibility criteria, since observational studies will be included in the search ?

Author response

We agree with the reviewer and have altered our references to PECOS from PICOS (within the text and in table 1).

2. Why did the authors use PRISMA statement instead of MOOSE statement to develop this protocol?

Author response

We refer to the PRISMA statement as it is well established and relevant to a range of study types. Furthermore the we used PRISMA-P statement for systematic review protocols to ensure that our protocol was developed according to appropriate guidance.

The reviewer does, however, raise an interesting point as the MOOSE checklist is more specific to meta-analyses of observational studies. In completing and reporting the findings of the final review, the guidance of the MOOSE checklist will be highly relevant. These guidelines are not contradictory. Having developed the protocol with reference to the PRISMA statement, we have retained this within the protocol. However, where a meta-analysis is undertaken, we will ensure that the MOOSE checklist is adhered to when reporting the findings. We have altered the text to reflect this.

“The review will be conducted and reported according to the Preferred Reporting Items for Systematic Reviews and Meta-Analyses (PRISMA) statement.(19) Where a meta-analysis is undertaken, we will report findings according to the Meta-analyses Of Observational Studies in Epidemiology checklist.”
Page 7 lines 139-142

3. Page 10: "studies should report the association with the outcome in the presence of absence of frailty". I think the authors would like to say "in the presence or absence of frailty".

Author response

Thank you for this comment. We have revised the section in question to clarify the meaning:

“For assessing the association between frailty and clinical outcomes in the context of diabetes, studies should report the association between frailty and the outcome of interest. This may be reported either as the association with the presence or absence of frailty (in the case of a binary or categorical measure) or the association between the degree of frailty and the outcome (in the case of a continuous or ordinal measure of frailty).” Page 10, lines 178-182

4. I think the authors should explain in more detail how the outcomes should be measured.

Author response

As we include a wide range of outcomes, the way in which outcomes are measured may potentially vary within, and between outcomes. At the data extraction stage, the method of analysis for any relevant outcome will be extracted. Following this, within each outcome, studies will be compared to judge the comparability of the measures used. We have expanded on the description of outcomes within data extraction to elaborate on this:

“As we include a wide range of outcomes, it is likely that the way outcomes are assessed will vary

depending on the outcome in question. Studies may also assess similar outcomes (e.g. hospital admission) in different ways (e.g. number of admissions over specified follow-up, time to first admission, presence of absence of admission during follow-up). For the outcomes listed above, we will extract data regardless of the method of assessment. Heterogeneity in the way outcome data are collected will be used to inform the approach to data synthesis (i.e. meta-analysis versus narrative synthesis). For each outcome reported we will record

- The method of outcome assessment (e.g. linkage to healthcare records, face-to-face assessment, questionnaire etc.)
- Method of analysis (e.g. time-to-event, mean difference etc.)
- The association between frailty and the outcome (e.g. prevalence, odds ratio, hazard ratio etc.)
- Adjustment for any potential confounders
- Length of follow-up over which the outcome was assessed
- Method of analysis of competing risks when assessing each outcome

Where available, we will also extract data on both relative (e.g. hazard ratios) and absolute (e.g. events per 1000 people) associations with outcomes.” Page 14-15, lines 264-280

5. Authors should review in their protocol a possible language bias.

Author response

Thank you for highlighting this. We have added the following text to the strengths and limitation section at the start of the manuscript:

“By including only English language articles, there is a chance of language bias in the results of the review.” Page 3, lines 66-67

We have also added the following text to the section describing the search strategy:

“This language restriction is a pragmatic decision, however we acknowledge that this may lead to a language bias in the results, potentially excluding relevant studies published in other languages.” Page 11, lines 200-202

6. Authors should include gray literature in the search.

Author response

We thank the reviewer for this suggestion. While we acknowledge that limiting our inclusion criteria to published literature risks publication bias, there is a balance between this and the time and resource required to undertake a full and comprehensive search for Grey literature. For the review, we have elected not to include Grey literature and to limit our review to published studies. The epidemiological literature on frailty is large and rapidly growing (as such additional time spent searching for Grey literature risks review findings becoming out of date). Also, as the majority of estimates on frailty prevalence in diabetes will be drawn from large population based cohort studies, and frailty prevalence is largely descriptive rather than being a ‘significant’ or ‘null’ result, we anticipate that most findings from large cohorts will be included within the published literature.

We acknowledge, however, that this is a limitation of our methodology, and have edited the manuscript to make this explicit. We have added the exclusion of Grey literature to the ‘other exclusions’ section of table 1. We have also added a bullet-point to the ‘Strengths and Limitations’ section which comes at the top of the manuscript stating:

“We exclude Grey literature, which may lead to publication bias.” Page 3, line 68

7. The methodological assessment of cross-sectional studies should be performed using another tool. See <https://www.nhlbi.nih.gov/health-topics/study-quality-assessment-tools>

Author response

Thank you for this comment. On considering this comment, we acknowledge there is a need for greater clarity on how cross-sectional studies are to be quality assessed. The link suggested by the reviewer does not contain a quality assessment tool specific to cross sectional studies (rather, a tool for observational cohort and cross-sectional studies, with some questions relevant only to longitudinal studies). This is a similar issue to the use of the Newcastle-Ottawa checklist, where not all questions are applicable to cross sectional studies. We have therefore adapted the checklist (based on previous adaptations for cross sectional studies) in such a way to allow us to consistently apply a quality assessment tool to all studies in the review.

Our review will assess 2 main aspects: prevalence and association with outcomes. Cross sectional studies may only assess prevalence, however some longitudinal studies may include both baseline prevalence and association with outcomes. We believe it will be important to apply the same quality assessment criteria to all prevalence studies (i.e. both cross-sectional studies and longitudinal studies reporting baseline prevalence). We also think it will be most straightforward to use only one quality assessment tool for each study.

We therefore have adapted the Newcastle-Ottawa scale whereby the first 5 questions will be applied to all studies (cross sectional and longitudinal) and are relevant to prevalence estimates, and the remaining 4 questions will be applied to longitudinal studies only (providing a fuller quality assessment for studies assessing outcomes). This will ensure comparability of quality assessment between studies assessing prevalence.

We have expanded on our description of the quality assessment to make this approach and the rationale clear.

“The Newcastle-Ottawa scale will be used to assess the risk of bias for each study.(21) This scale is widely used for the assessment of observational studies, and has frequently been adapted to the context of specific systematic reviews. We have adapted the criteria in order to be explicit about how the ‘exposure assessment’ related to frailty: specifically, awarding one point for the use of a validated frailty assessment measure. For cross-sectional studies, only the first 5 elements of the scale were relevant to quality assessment (the remainder concerning the longitudinal assessment of outcomes). We will use this subsection of the Newcastle-Ottawa scale to assess the quality of cross-sectional studies to allow direct comparability with the baseline assessments of longitudinal studies (from which we will also extract data on frailty prevalence).” Page 15, lines 282-292

We have also added the quality assessment form as a supplementary appendix.

8. The strategies for planning the meta-analysis are vague. Please describe in detail how the measures of proportion and association can be summarized. Will there be sensitivity analysis and publication bias analysis?

Author response

Thank you for this comment. We have added the following text to be more explicit about how we will assess whether to perform a meta-analysis to summarise the data:

“With regards to the prevalence of frailty, different frailty measures will be considered separately (i.e. we will not perform a meta-analysis of frailty prevalence measured using different scales). We will also consider community studies separately from studies focussing on outpatient clinic populations (as

these may represent people with more severe diabetes), inpatients or people living in residential care. We will also assess the inclusion criteria and demographics of the sample population, with particular attention to age (as frailty is strongly associated with age) and sex (as women tend to have a higher prevalence of frailty than men) to determine the most appropriate method of synthesis. Where samples have been drawn from populations with a markedly different age/sex structure, a pooled estimate of the mean prevalence of frailty across these studies is unlikely to be a meaningful summary. Similarly, other inclusion criteria used by the individual studies (such as excluding 'institutionalised' people, people with cognitive impairment, of people with impaired mobility unable to attend an assessment) may disproportionately impact on the estimation of frailty prevalence. The appropriateness, or otherwise, of a meta-analysis of frailty prevalence will be judged only after examination of these aspects of the included studies.

For the assessment of outcomes, the approach to synthesis will also be judged based on heterogeneity of the method of outcome assessment and the analytic approach. As above, different frailty measures will be considered separately.” Page 16, lines 297-313

Finally, we have added:

“We will use funnel plots to assess for potential publication bias.” Page 17, lines 321-322

VERSION 2 – REVIEW

REVIEWER	Nanna Lindekilde University of Southern Denmark, Denmark
REVIEW RETURNED	05-May-2020

GENERAL COMMENTS	Thank you for the opportunity to review this resubmitted version. The authors have responded sufficient to my questions and made corrections and elaborations when needed. I only have a minor comment before I can recommend publication of this paper: During the paper, the authors state that they will include people with diabetes (type1, type 2, or mixed/unspecified). However, in page 9, L 147, they state: “We will include studies analysing data from people with any form of diabetes.”. This discrepancy must be corrected. In the search terms the authors have also included terms for additional types of diabetes e.g. MODY, which support the solution of including people with any form of diabetes. Likewise, I would also recommend the authors to include people with all type of diabetes. If they choose not to, an argument for exclusion of some types of diabetes is needed. When the authors have corrected this element throughout the paper, I recommend publication of this paper.
--

VERSION 2 – AUTHOR RESPONSE

Reviewer: 1

Thank you for the opportunity to review this resubmitted version. The authors have responded sufficient to my questions and made corrections and elaborations when needed.

Author response:

Thank you for this comment.

Reviewer comment:

I only have a minor comment before I can recommend publication of this paper:

During the paper, the authors state that they will include people with diabetes (type1, type 2, or mixed/unspecified). However, in page 9, L 147, they state: "We will include studies analysing data from people with any form of diabetes.". This discrepancy must be corrected. In the search terms the authors have also included terms for additional types of diabetes e.g. MODY, which support the solution of including people with any form of diabetes. Likewise, I would also recommend the authors to include people with all type of diabetes. If they choose not to, an argument for exclusion of some types of diabetes is needed.

Thank you for this comment. We are grateful to the reviewer for highlighting this inconsistency. We have made the following changes to the manuscript to address this issue:

"Adults with any type of diabetes mellitus" – line 46 – abstract

"Identify which frailty measures have been used to assess frailty in people with diabetes (any type, including mixed/unspecified)" – lines 136-137 – introduction.

"Diabetes (any type, including mixed or unspecified)" – table 1

"We will therefore include any study which includes people with any type of diabetes (including type 1, type 2 diabetes, secondary or monogenic diabetes, or people with unspecified diabetes)." Lines 157-158

Deleted "type 1 or type 2" from lines 240